# Simulating the Evolution of von Neumann Entropy in Black Hole Hawking Radiation Using Biphoton Entanglement

**DOI:** 10.3390/e27030236

**Published:** 2025-02-25

**Authors:** Zhuoying Li, Haoshen Fan, Xingwen Zhao, Qinfei Wu, Ji Bian, Yang Liu, Le Luo

**Affiliations:** 1School of Physics and Astronomy, Sun Yat-Sen University, Zhuhai 519082, China; lizhy379@mail2.sysu.edu.cn (Z.L.); fanhsh5@mail2.sysu.edu.cn (H.F.); zhaoxw@mail.ustc.edu.cn (X.Z.); wuqf8@mail.sysu.edu.cn (Q.W.); bianji@ustc.edu.cn (J.B.); 2Quantum Science Center of Guangdong-HongKong-Macao Greater Bay Area, Shenzhen 518048, China; 3Shenzhen Research Institute, Sun Yat-Sen University, Shenzhen 518057, China; 4Guangdong Provincial Key Laboratory of Quantum Metrology and Sensing, Sun Yat-Sen University, Zhuhai 519082, China; 5State Key Laboratory of Optoelectronic Materials and Technologies, Sun Yat-Sen University, Guangzhou 510275, China

**Keywords:** black hole information paradox, Page curve, two-photon entanglement, tomography, von Neumann entropy

## Abstract

Addressing the black hole information paradox necessitates the exploration of various hypotheses and theoretical frameworks. Among these, the proposition to utilize quantum entanglement, as introduced by Don N. Page, shows great promise. This study builds upon Page’s theoretical foundation and proposes a simplified model for elucidating the evolution of black hole von Neumann entropy. This model simulates the process of Hawking radiation using entangled photon pairs. Our experiment suggests that quantum entanglement may offer a plausible avenue for resolving the paradox, thereby lending support to Page’s proposal. The results suggest that this model may contribution to the exploration of one of the most profound puzzles in theoretical physics.

## 1. Introduction

The space–time curvature [1] and the no-hair theorem of black holes [2] indicate that any matter crossing the event horizon becomes irretrievably trapped, with its information disappearing. Only three properties are retained by the black hole: mass, charge, and angular momentum. This characteristic, however, appears to conflict with the second law of thermodynamics, as the infall of high-entropy matter into a black hole would seemingly result in a net decrease in the entropy of the universe. Jacob D. Bekenstein addressed this paradox by proposing that black holes have entropy, which is directly proportional to the surface area of their event horizon [3,4]. Meanwhile, Stephen Hawking expanded on this concept by introducing the theory of black hole radiation, which demonstrated that black holes emit thermal radiation [5]. Although Hawking radiation has not been directly observed, recent progress in using the Bose–Einstein condensate of ultra-cold atoms to simulate acoustic black holes has verified Hawking radiation to some extent [6], preliminarily affirming the correctness of Hawking’s theory of radiation. Hawking radiation implies that evaporating black holes emit purely thermal radiation, which carries no quantum information. If a black hole undergoes complete evaporation, the information associated with the matter that entered it would be irretrievably lost. This outcome conflicts with the fundamental principle of information conservation in quantum mechanics, leading to what is known as the black hole information paradox. The black hole information paradox reveals the potential conflict between general relativity and quantum mechanics under extreme gravitational conditions, involving fundamental issues such as quantum information conservation, black hole thermodynamics, and the second law of thermodynamics. Resolving this paradox not only constitutes a crucial test of black hole thermodynamics and the principle of quantum information conservation but also carries far-reaching implications for exploring new physical theories, deepening our understanding of the fundamental laws of the universe, and developing novel observational and experimental methodologies. Don N. Page proposed that quantum information is encoded in the correlation between radiation particles and the black hole [7,8]. He presented the general evolution process of the radiation entanglement entropy, namely the Page curve. Recently, researchers achieved a significant breakthrough by calculating the Page curve using semi-classical techniques, which was accomplished by investigating black holes in asymptotically anti-de Sitter spacetimes connected to a conformal field theory reservoir [9,10]. Moreover, the Island rule formula of Hawking radiation fine entropy was put forward, and the Page curve was derived from the angle of pure gravity [11,12,13]. The challenge of directly observing Hawking radiation makes it difficult to experimentally validate Hawking’s theory. However, recent advancements in gravity and black hole analogues—such as Bose–Einstein condensates [6], optical systems [14,15], and water waves [16]—have provided new insights into black hole dynamics. The aim of this paper is to demonstrate the time-dependent change in entropy of Hawking radiation using a simple and easily verifiable model. This study builds upon Page’s theory by modeling the black hole as a two-photon entangled system. Through a direct calculation of the von Neumann entropy, we derived and experimentally verified the Page curve, illustrating the time evolution of black hole entropy, which initially increases and subsequently decreases back to its initial value during Hawking radiation, thus successfully reproducing the key qualitative features of black hole evaporation and suggesting that unitarity may be preserved in physical black holes. Although the simplified model does not fully capture the complex dynamics of real black holes and has certain limitations in addressing unknown physical phenomena, it offers a valuable perspective for intuitively understanding the black hole information paradox and the process of information transfer, thereby laying a foundation for further research in this field.

## 2. Theory

The solution proposed by Don N. Page regarding Hawking radiation and quantum entanglement suggests that maximally entangled pairs of particles can be viewed as being in a pure state as a whole [17]. However, when this pair of particles is separated, each individual particle will be in a completely mixed state, possessing maximal von Neumann entropy. Vacuum quantum fluctuations near the event horizon of a black hole produce pairs of virtual particles in a maximally entangled state. Under the influence of the black hole’s intense gravitational field, one particle from each pair is captured by the black hole, while the other escapes, transitioning to a real particle and forming what is observed as Hawking radiation. By assuming that the increase in von Neumann entropy of the black hole system arises exclusively from the particle evolution process between the black hole and its Hawking radiation, without external influences, it follows that the maximum von Neumann entropy of the combined black hole and radiation system must be equal to each other. Furthermore, this maximum entropy should not exceed the semi-classical entropy of the system, ensuring consistency with established thermodynamic principles [18].(1)$$\begin{array}{cc} S_{vN} & =\min\left[{\tilde{S}}_{BH},{\tilde{S}}_{\mathrm{rad}\;}\right]\\ \; & =4\pi M_{0}^{2}\cdot \min\left\{{\left(1-t/t_{\mathrm{decay}\;}\right)}^{2/3},1.48472\left[1-{\left(1-t/t_{\mathrm{decay}\;}\right)}^{2/3}\right]\right\} \end{array}$$

${\tilde{S}}_{BH}$ and ${\tilde{S}}_{\mathrm{rad}\;}$ represent the semi-classical entropies of the black hole and the Hawking radiation, respectively. $M_{0}$ denotes the initial mass of the black hole at time *t* = 0, while $t_{\mathrm{decay}\;}=8895\;M_{0}^{3}$ specifies the decay time for a non-rotating uncharged black hole. The 1.48472 is the ratio of the increasing entropy of the radiating part to the decreasing entropy of the black hole part.

Based on this theory, this paper views the black hole as a system with quantum entanglement properties and directly calculates the evolution of von Neumann entropy over time during the Hawking radiation process using a biphoton entanglement model, without relying on semi-classical entropy calculations. We construct a simplified model of Hawking radiation by assuming a non-rotating uncharged Schwarzschild black hole in flat spacetime. In this model, the event horizon is treated as the radiating surface, emitting photons that follow a blackbody radiation distribution. The black hole evaporation process and Hawking radiation are equated to the gradual separation of N maximally entangled photon pairs. In this process, it is assumed that the radiation preferentially separates unsplit photon pairs until each photon inside the black hole has its entangled partner in the Hawking radiation. The moment when this condition is reached can be considered equivalent to the Page time.

According to the von Neumann entropy formula for black holes given in Ref. [19] (see Equation (6.2)),(2)$$S={min}_{X}\left\{{\mathrm{ext}}_{X}\left[\frac{\mathrm{Area}(X)}{4G_{N}}+S_{\text{simi-cl}}\left(\Sigma_{X}\right)\right]\right\}.$$

We can qualitatively explain the rationality of the ‘splitting’ assumption. From this formula, the rationality of the ‘splitting’ assumption can be qualitatively explained. From this formulation, it is evident that the total entropy of the black hole is composed of two distinct components. The first component, $\frac{\mathrm{Area}(X)}{4G_{N}}$, is associated with the geometric properties of the black hole, commonly understood as its information storage capacity. Therefore, the surface area of the selected surface *X* can be considered as the first contribution to the generalized entropy. The second part $S_{\text{simi-cl}}\left(\Sigma_{X}\right)$ involves quantum entanglement properties, describing the interaction between quantum fields. In the current model, since external factors are not considered, the second contribution comes from the vacuum quantum fields in the region $\Sigma_{X}$, which consist of particle pairs generated by vacuum fluctuations. By adjusting the position of surface *X* and searching for the extremum of the generalized entropy and selecting the smallest value, namely ${min}_{X}\left\{{\mathrm{ext}}_{X}[]\right\}$ (‘ext’ represents taking the extremum of the generalized entropy, and ‘min’ indicates that when there are multiple extremal surfaces, the one that minimizes S is chosen), the von Neumann entropy of the black hole can then be determined. Consequently, a simplified depiction of the black hole structure, illustrated in Figure 1, can be constructed. In the figures, the red solid line is located a few Schwarzschild radii away from the singularity and is defined as the boundary of observation; the blue solid line represents the event horizon, situated between the singularity and the boundary. The position of surface *X* can be freely chosen between the singularity and the event horizon, with the region enclosed between surface *X* and the boundary designated as $\Sigma_{X}$.

Before the black hole begins emitting Hawking radiation, the vacuum quantum field does not contribute to the entropy, and there are no isolated particles in $\Sigma_{X}$. At this stage, the minimum von Neumann entropy corresponds to selecting surface *X* directly at the singularity, with the von Neumann entropy being zero. Therefore, we may initially fix surface *X* at the singularity (as shown in Figure 1b). When Hawking radiation commences, pairs of entangled particles at the event horizon become separated and are emitted in opposite directions, with some particles gradually crossing the boundary and escaping from the black hole, while the correlated entangled particles contained in the region $\Sigma_{X}$ move inward. At this point, the von Neumann entropy inside and outside the black hole increases simultaneously and in equal measure. We define this evolution mode as Mode 1. However, as the radiation process continues, the von Neumann entropy corresponding to surface *X* placed at the singularity may no longer be the minimum. In this case, surface *X* can be moved to the event horizon (as shown in Figure 1c), where it becomes evident that $\Sigma_{X}$ contains only a few radiated particles; so, its von Neumann entropy is not zero. On this basis, if surface *X* is moved further toward the singularity (as shown in Figure 1d), the value of the geometric term decreases, and since it includes entangled pairs of radiated particles, the second term also decreases. Ultimately, the generalized entropy of the entire system decreases and reaches a minimum. We define this evolution mode as Mode 2.

It is worth noting that in Mode 2, the minimum value corresponding to surface *X* is always near the event horizon, and this minimum closely corresponds to the semi-classical entropy of the black hole, commonly referred to as the area entropy. With the progression of the radiation and the evaporation of the black hole, these entropies gradually decrease. Accordingly, the von Neumann entropy of the black hole corresponds to the extremum value obtained from the two previously described modes: the result of Mode 1 starts from zero and continuously increases as radiation proceeds, while the result of Mode 2 gradually decreases to zero, as the black hole evaporates. Taking the minimum of these two values reveals a trend that the von Neumann entropy first increases and then decreases, providing insight into the black hole information paradox. This argument corresponds directly to the ‘splitting’ hypothesis of biphoton entanglement (as shown in Table 1), demonstrating the rationality of the proposed hypothesis.

Returning to the simplified black hole radiation model mentioned earlier, in order to calculate the evolution of the number of radiated particles over time, we use the Unruh Effect to compute the Hawking temperature of the black hole [20], treating the event horizon $\left(r_{0}=2M\right)$ as the radiating surface. From this, we obtain the total energy and total number of particles radiated per unit time [more details in Appendix A]:(3)$$T\left(\infty \right)=\frac{1}{8\pi M}$$(4)$$\begin{array}{cc} \; & P=4\pi r_{0}^{2}\cdot p=C_{p}M^{-2}\\ \; & N=4\pi r_{0}^{2}\cdot n=C_{N}M^{-1}. \end{array}$$The model assumes that the mass loss entirely results from photons radiated outward from the event horizon; thus, according to the mass–energy equation, we can derive the following:(5)$$P=\frac{\Delta E}{\Delta t}=\frac{-\Delta Mc^{2}}{\Delta t}=-\frac{dM}{dt}=C_{p}M^{-2}.$$

Solving this differential equation yields the relationship between mass and time evolution:(6)$$M\left(t\right)=M_{0}{\left(1-t/\alpha \right)}^{1/3}$$(7)$$\alpha =\frac{M_{0}^{3}}{3C_{p}}=5120\pi^{2}M_{0}^{3}.$$

By incorporating the temporal evolution of mass into the expression for the particle number, it can be determined that the variation in the number of particles emitted per unit time over time is(8)$$N\left(t\right)=C_{N}M_{0}^{-1}{\left(1-t/\alpha \right)}^{-1/3}.$$

Through integrating the above formula with respect to time, we can obtain that the variation of the total particle number of the radiative portion at time *t* with respect to time is(9)$$N_{\mathrm{radiation}\;}\left(t\right)=\frac{3}{2}\alpha C_{N}M_{0}^{-1}\left(1-{\left(1-t/\alpha \right)}^{2/3}\right).$$

The complete evolution of von Neumann entropy over time is(10)$$\begin{array}{cc} S_{vN} & =\frac{3}{2}\alpha C_{N}M_{0}^{-1}\cdot \min\left[\left(1-{\left(1-t/\alpha \right)}^{2/3}\right),{\left(1-t/\alpha \right)}^{2/3}\right]\cdot \ln2\\ \; & \approx 3.22464M_{0}^{2}\cdot \min\left[\left(1-{\left(1-t/\alpha \right)}^{2/3}\right),{\left(1-t/\alpha \right)}^{2/3}\right]. \end{array}$$

It is particularly noteworthy that although this model is based on discrete photon pairs, the derived particle number formula (Equation (Equation 9)) exhibits qualitative consistency with the semi-classical entropy calculation (Equation (Equation 1)). After the Page time, the entropy decrease in both cases originates from the complete separation of entangled pairs. This result indicates that while the discrete model serves as a simplified framework, it effectively captures the universal principles of information transfer in Hawking radiation.

## 3. Experiment

Based on the aforementioned theory, we designed the experimental optical setup shown in Figure 2 to verify the biphoton entanglement scheme. As shown in Figure 2a, a pump laser with a wavelength of 405 nm was directed through two half-wave plates (HWP) and two quarter-wave plates (QWP) to precisely adjust and control its polarization state. The adjusted laser then entered a BBO (Beta Barium Borate) crystal, generating entangled photon pairs through the spontaneous parametric down-conversion (SPDC) process. The laser was a 40 mW continuous wave laser with output polarized in the vertical direction. The BBO crystal consisted of two thin BBO crystals with their optical axes perpendicular to each other, bonded together for the type-I parametric down-conversion process. The generated entangled photon pairs passed through a half-wave plate, which adjusted the polarization of each optical path and were transmitted via optical fibers to two independent paths, A and B. Figure 2b illustrates the detailed structure of the two entanglement test paths, A and B. Each path contained a QWP and two HWPs to precisely adjust the polarization state of the photons. A polarizer (PP) was used to select photons with specific polarization states, and an optical coupler (OC) ensured the efficient transmission of the optical signal. The entangled photon pairs entered path A and path B, respectively. After being adjusted by the corresponding optical components, they finally reached single-photon detectors (SPD-A and SPD-B). A coincidence counter (CCI) was employed to evaluate whether the photon detection results from both paths satisfied the coincidence conditions, thus confirming the entangled state of the biphotons.

In this experiment, the black hole was simplified to a system composed of five pairs of maximally entangled photons ($A_{1}$–$B_{1}$, $A_{2}$–$B_{2}$, ……, $A_{5}$–$B_{5}$), generated via SPDC. The black hole evaporation process was entirely framed within the context of quantum optics theory, with the “horizon”—treated as a phenomenological construct corresponding to the optional surface *X*-serves as an operational partitioning tool. This tool divided the observable radiation (the extracted photons) from the simulated black hole microstates (the remaining photons) to characterize the entanglement dynamics between the radiated and residual components. Therefore, the evaporation process was mapped to the gradual evolution of these entangled photon pairs, with data selection performed in accordance with Page’s theory. At five distinct time intervals, we recorded real-time single-photon counts from SPD-A and real-time coincidence counts from the coincidence counter (see Appendix C for details). Among these data, the real-time single-photon counts of SPD-A represented the photons that preferentially escape from the black hole at the initial stage of the Hawking radiation simulation, thereby modeling the “thermal” nature of early black hole radiation. During this stage, the radiation did not carry the black hole’s information, and the number of radiated photons continued to increase until it reached a maximum (five sets of single-photon counts $A_{1}$∼$A_{5}$, corresponding to a 50% loss in black hole mass). Subsequently, photons traveling along path B ($B_{i}$) began to be emitted, simulating the transfer of black hole information to the radiation. In this phase, part of the radiation reconstructed the information correlations via entanglement. This reconstruction was evidenced by the average coincidence count—indicating entanglement reestablishment—and an entanglement fidelity exceeding F > 0.977, which strongly suggest that the process was consistent with unitarity, as no irreversible loss of information was detected within the measured subsystem. As evaporation proceeded, all photons were ultimately radiated away.

By adjusting the half-wave plates and quarter-wave plates to different angles, we changed the measurement basis and used the photon counts in the current basis vectors as direct measurement values to reconstruct the system’s density matrix. In the experiment, H and V corresponded to the horizontal and vertical polarizations, A and D corresponded to the 45° and −45° diagonal polarizations, and L and R corresponded to the left-hand and right-hand circular polarizations. We used different polarizations as basis vectors to measure the single-photon counts in a single channel to characterize the behavior of individual photons. For entangled photon pairs, we selected H, V, D, and R as the bases and constructed 16 basis vector combinations. By recording the coincidence counts in both channels, we reconstructed the density matrix of the biphoton system.

## 4. Results

The density matrix of a single photon can be expressed in its expanded form as [21](11)$$\hat{\rho }=\frac{1}{2}\sum_{i=0}^{3}S_{i}{\hat{\sigma }}_{i},$$
where $\sigma_{0}$ represents the identity matrix, and the remaining three are in the order of Pauli x, y, z matrices. The parameter $S_{i}$ is called the Stokes parameter and represents the components of the density matrix on the individual basis vectors, which can be defined by three sets of generally orthogonal measurement bases:(12)$$\begin{array}{cc} \; & S_{0}=P_{|H\rangle }+P_{|V\rangle }\\ \; & S_{1}=P_{|D\rangle }-P_{|A\rangle }\\ \; & S_{2}=P_{|R\rangle }-P_{|L\rangle }\\ \; & S_{3}=P_{|H\rangle }-P_{|V\rangle }. \end{array}$$The P here refers to the probability associated with the corresponding measurement basis. For example, the probability for the horizontal polarization state |H〉(or |D〉) is given by $P_{|H\rangle }=\frac{n_{|H\rangle }}{n_{|H\rangle }+n_{|V\rangle }}\left(P_{|D\rangle }=\frac{n_{|D\rangle }}{n_{|D\rangle }+n_{|A\rangle }}\right)$, where $n_{|H\rangle }\left(n_{|D\rangle }\right)$ and $n_{|V\rangle }\left(n_{|A\rangle }\right)$ represent the number of photons measured in the |H〉(|D〉) and |V〉(|A〉) states, respectively. For more details, refer to Appendix C. By measuring the photon counts under the measurement bases H, V, D, A, and L, R and processing the data accordingly, the Stokes parameters can be obtained, which can then be used to reconstruct the density matrix of a single photon $\rho_{p1}$∼$\rho_{p5}$.

For entangled photon pairs, the density matrix is represented as a 4 × 4 matrix. In the space of 4 × 4 Hermitian matrices, a total of 16 basis vectors are required. These basis vectors can be constructed using the pairwise Kronecker product of four Pauli matrices.(13)$$\hat{\rho }=\frac{1}{4}\sum_{i,j=0}^{3}S_{ij}\left(\sigma_{i}⊗\sigma_{j}\right)$$

The Stokes parameter is satisfied [21]:(14)$$S_{ij}=\mathrm{Tr}\left(\hat{\rho }\left(\sigma_{i}⊗\sigma_{j}\right)\right).$$

When using different measurement vectors to measure the quantum state, the measurement probability value meets(15)$$P_{s}=\langle \psi_{s}\left|\rho \right|\psi_{s}\rangle =\frac{1}{4}\sum_{i,j=0}^{3}\langle \psi_{s}\left|S_{ij}\left(\sigma_{i}⊗\sigma_{j}\right)\right|\psi_{s}\rangle .$$$\psi_{s}\left(s=1∼16\right)$ is a selected set of sixteen measurement vectors. The detailed calculation process of the measurement probability $P_{s}$ (For example, $P_{|HH\rangle }$) can be found in Appendix C.

After measuring the quantum state with 16 measurement vectors, Stokes parameters are written in the matrix form $\hat{S}$ with(16)$$\hat{P_{s}}=\frac{1}{4}\hat{T}\hat{S},$$
where the matrix entries of a *T* matrix are is $T_{ij}=\langle \psi_{i}\left|\left(\Gamma_{j}\right)\right|\psi_{i}\rangle \left(i,j=1,2,3,\ldots ,16\right)$, with $\Gamma_{1∼16}=\sigma_{0∼3}⊗\sigma_{0∼3}$. $\hat{P_{s}}$ and $\hat{S}$ are column matrices composed of 16 probability values and Stokes parameters.

According to the above formula, the matrix $\hat{S}$ is $\hat{S}=4{\hat{T}}^{-1}\hat{P_{s}}$. The Stokes parameters are subsequently calculated and substituted into Equation (Equation 13) to reconstruct the density matrix. However, tomographic measurements of density matrices can occasionally yield results that violate fundamental properties, such as positivity. Specifically, if the density matrix exhibits negative eigenvalues, it becomes impossible to compute the von Neumann entropy. To address this issue, the maximum likelihood estimation (MLE) method [22] is employed to optimize the density matrix $\rho_{e6∼e10}$. For a detailed explanation of the procedure, please refer to Appendix D.

The density matrices of the single-photon state and two-photon state are shown in Figure 3a and Figure 3b, respectively. In the two-photon density matrix, the real part is the MLE-density matrix, and the virtual part is the raw density matrix. As Hawking radiation progresses, the radiation particles will increase with time, and the density matrix of the first five particles can be expressed as(17)$$\rho_{n}=\rho_{p1}⊗\rho_{p2}\ldots ⊗\rho_{pn}.$$Since the first five photons emitted are independent single photons without entanglement, the density matrix at this time is the direct product of the single-photon density matrices. Starting from the radiation of the sixth photon, entangled photon pairs will appear in the radiation section, giving(18)$$\begin{array}{cc} \rho_{6}= & \rho_{e6}⊗\rho_{p2}⊗\rho_{p3}⊗\rho_{p4}⊗\rho_{p5}\\ \rho_{7}= & \rho_{e6}⊗\rho_{e7}⊗\rho_{p3}⊗\rho_{p4}⊗\rho_{p5}\\ \; & \ldots \\ \rho_{10}= & \rho_{e6}⊗\rho_{e7}⊗\rho_{e8}⊗\rho_{e9}⊗\rho_{e10}. \end{array}$$

**Figure 3 entropy-27-00236-f003:**
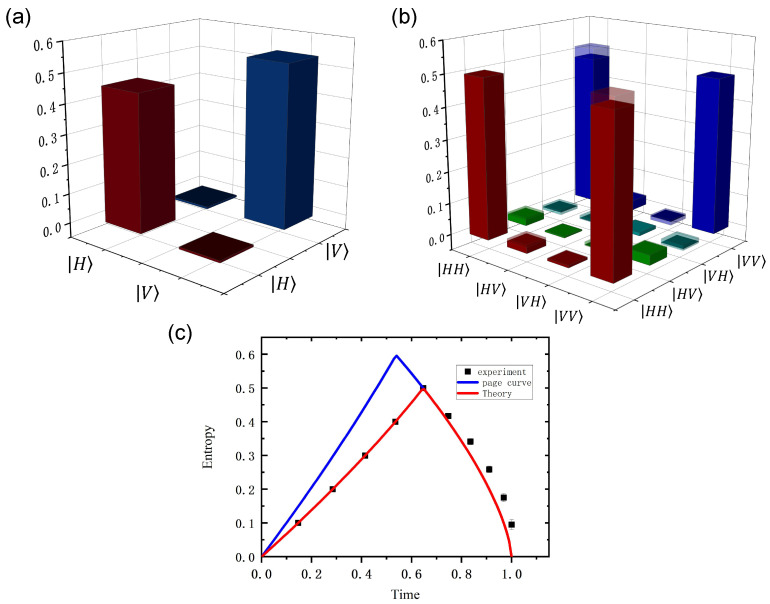
(**a**) The real part of the single-photon state density matrix. (**b**) The real part of the two-photon state density matrix.The semi-transparent region shows the difference between the raw density matrix and the MLE-reconstructed density matrix. (**c**) Evolution of von Neumann entropy over time. The solid red line represents the theoretical prediction (Equation (Equation 10)), the solid blue line corresponds to the Page curve (Equation (Equation 1)), and the black dots indicate the experimental results. To facilitate a clearer comparison of the von Neumann entropy trends, the entropy values have been normalized.

By substituting these ten density matrices of the radiation section into the formula $S_{vN}=-\mathrm{Tr}\left[\hat{\rho }log\hat{\rho }\right]$, the corresponding von Neumann entropy can be calculated. Then, according to Equation (Equation 9), the corresponding time of the current partial particle number of radiation can be inversely solved, and the change in the von Neumann entropy with time in the process of Hawking radiation can be obtained.

The von Neumann entropy curve plotted from the experimental results is highly consistent with the theoretically predicted curve (as shown in Figure 3c). This consistency demonstrates the reliability of the theoretical derivation and also validates the feasibility of the experimental setup. The result also indicates that during the black hole evaporation process, the quantum information is initially stored in the black. The trend of entropy increasing and then decreasing in the hole is gradually transferred to the radiation through entangled photon pairs in Hawking radiation, which directly reflects the dynamic process of information transfer from the black hole to the radiation.

Entropy increase phase: The internal information of the black hole appears to be “lost” due to the partial separation of entangled photon pairs.

Entropy decrease phase: After the Page time (t≈0.65), the radiation part contains the complete information of the entangled pairs, restoring information conservation.

This result is in agreement with Page’s prediction, which indicates that the process preserves unitarity and also suggests that the dynamic separation mechanism of entangled photon pairs can effectively simulate the resolution of the black hole information paradox.

Since the biphoton entanglement model proposed in this paper cannot consider the scattering effects included in the radiation process of the Page model, the ‘transition time’ of the two curves does not completely coincide. Nevertheless, the similar trend exhibited by both curves is sufficient to demonstrate the validity of this model and further supports the reliability of resolving the black hole information paradox through entanglement theory.

Meanwhile, it is worth noting that there are some experimental errors in the figure. These errors mainly originate from the limitations in the stability of the biphoton entanglement source, as well as systematic errors in the single-photon detectors and optical components. Additionally, fluctuations in the environmental conditions during the experiment also had some impact on the results. Although these errors did not significantly affect the accuracy of the overall trend, it is necessary to further improve the stability of the entanglement source, reduce systematic errors in the detection equipment and optical components, and enhance control over environmental conditions in future experiments to improve the precision and repeatability of the experimental results.

## 5. Conclusions

This paper presented a simulation of the evolution of von Neumann entropy during Hawking radiation using a system of entangled photon pairs. The information paradox arises from the apparent contradiction between the purportedly thermal characteristics of Hawking radiation and the fundamental principle of quantum information conservation. Our experimental framework revealed that through controlled spatial separation and subsequent interference of entangled photon pairs, the quantum state of the radiation subsystem undergoes a transformation sequence, starting from an initial pure state (characterized by low entropy) to a mixed state (exhibiting elevated entropy) and ultimately recovering purity (returning to low entropy). This observation suggests that radiation possessing thermal-like properties may maintain quantum correlations with a black hole’s interior particles. Following complete spatial separation and subsequent recombination of all entangled pairs, the radiation subsystem’s von Neumann entropy approaches the initial value, indicating full information retention during the simulated black hole evaporation process. Therefore, the evolution is consistent with quantum mechanical unitarity principles and can be interpreted as the purification of the localized state via the leakage of photons, whih is central to addressing the black hole information paradox. However, since this evolution process does not provide a comprehensive and detailed simulation of the complete Hawking radiation process, its result should not be regarded as a definitive resolution of the black hole information paradox. To fully resolve the paradox, a more advanced experimental system and more rigorous theoretical support remain necessary. Nonetheless, this experiment clearly indicates that both the transfer of information and the reconstruction of entanglement during the process of black hole Hawking radiation can be effectively explained by fully quantum processes. This result not only deepens our understanding of the mechanisms underlying information conservation in black hole Hawking radiation but also indicates that entangled photon systems can serve as a promising platform for further exploring the fundamental processes in black hole thermodynamics.

In future research, we plan to incorporate an ion–photon entanglement system to further expand the current experimental framework, thereby simulating more complex black hole physics scenarios such as rotating (Kerr) black holes [23], charged (Reissner–Nordstrom) black holes [24], and quantum gravity effects [25,26] under higher-dimensional or dynamically evolving spacetimes, etc. In this ion–photon approach, the trapped ions can serve as the “black hole”, while the photons simulate the “Hawking radiation”. Their respective decoherence and entanglement recovery processes correspond to the apparent information loss and eventual restoration that occur during black hole evaporation. By encoding orbital angular momentum states or introducing effective charge interactions, it becomes possible to model the evolution of spin in a rotating black hole or charge in a charged black hole throughout the evaporation process, and to observe the dynamic transfer of quantum information between the radiated photons and the remaining system. Moreover, by combining high-dimensional entangled-state preparation, multi-modal quantum measurement techniques, and controllable noise channels, one can systematically investigate the impact of quantum fluctuations and spacetime quantization effects (e.g., “spacetime foam”) on information fidelity. The multi-degree-of-freedom controllability and high-precision manipulation afforded by ion–photon systems provide a highly flexible route for simulating the spin, charge, and complex spacetime structures of realistic black holes, offering valuable insights into the mechanisms of Hawking radiation and the black hole information paradox.

## Figures and Tables

**Figure 1 entropy-27-00236-f001:**
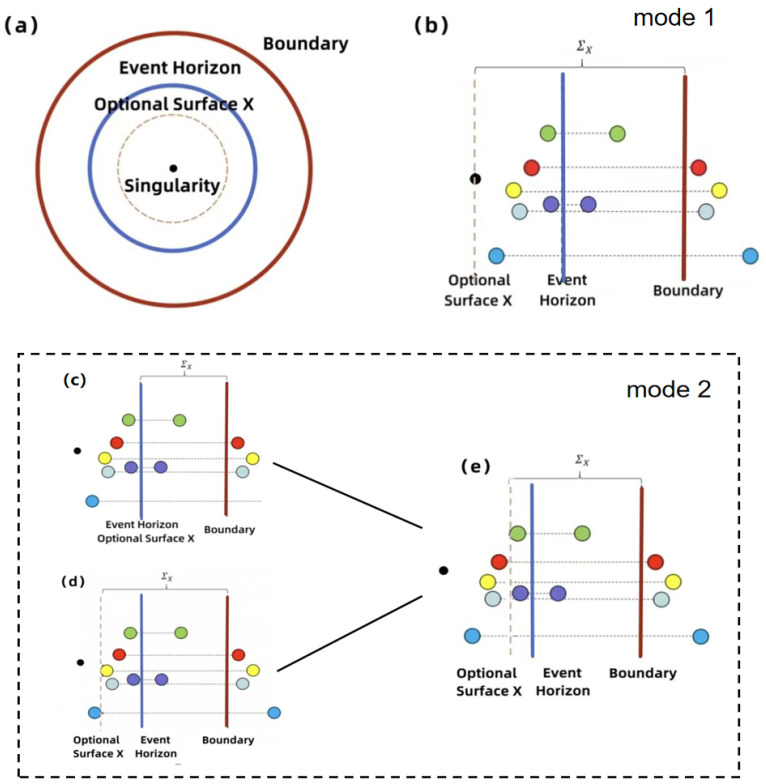
(**a**) Black hole schematic diagram. The red curve and blue curve represent the defined boundary of the black hole and the event horizon, respectively, while the dotted line denotes the optional surface *X*. In this context, the black hole region is characterized as the area confined within the defined boundary. The surface *X* position is adjustable, and the part between the surface *X* and the boundary is referred to as $\Sigma_{X}$. (**b**) Schematic diagram of Mode 1, where the optional surface *X* is located at a singularity. The balls of the same color connected by dotted lines represent a pair of positive and negative particles produced in the vacuum, and each such pair of particles is separated by the event horizon of a black hole. (**c**–**e**) Schematic diagram of Mode 2, where the optional surface *X* is not located at a singularity. (**c**) The optional surface *X* is directly selected at the event horizon. (**d**) The optional surface *X* is too close to the singularity. (**e**) The optional surface is selected at an appropriate position.

**Figure 2 entropy-27-00236-f002:**
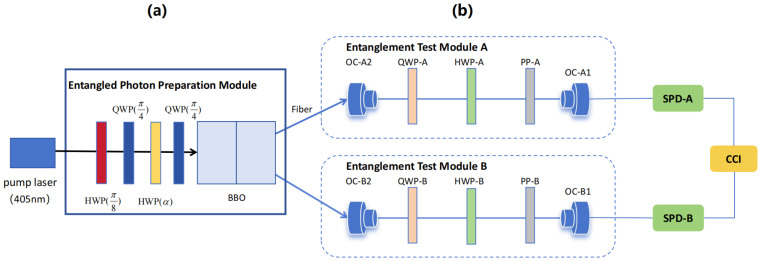
Experimental light path diagram: the left module is the optical path diagram of relative phase modulation and entangled state preparation, and the right module is the entanglement detection optical path diagram. (**a**) The pump laser (405 nm) is adjusted for polarization using wave plates before entering the BBO crystal to generate entangled photon pairs, which are then transmitted to paths A and B. (**b**) Each path contains a QWP and HWP for entanglement testing. A polarizer (PP) selects specific polarization states, and an optical coupler (OC) ensures signal transmission. The entangled photons eventually reach the single-photon detectors (SPD-A and SPD-B), with a coincidence counter (CCI) used to verify the photon entangled state.

**Table 1 entropy-27-00236-t001:** This is a comparison of the two methods.

Two-Photon Model Method	Minimizing Generalized Entropy Method
The von Neumann entropy of the radiation component	Mode 1
The remaining von Neumann entropy of a black hole	Mode 2 takes the extreme value
The black hole first emits one of the entangled photons	The vacuum fluctuation causes the radiation particles in the entangled pair to cross the cutoff surface
All entangled photon pairs have been separated	The results of mode 1 and mode 2 are equal
The radiation portion begins to fully contain entangled photon pairs	How to take the extreme value of mode 2

## Data Availability

The data that support the findings of this study are available from the corresponding author upon request.

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
