# Peer review of "Simulating the Evolution of von Neumann Entropy in Black Hole Hawking Radiation Using Biphoton Entanglement"

_entropy, 2025, doi:10.3390/e27030236_

Round 1

Reviewer 1 Report

Comments and Suggestions for Authors

The manuscript investigates the black hole information paradox through an experimental
approach, using biphoton entanglement to simulate Hawking radiation. By leveraging the von
Neumann entropy dynamics, the authors demonstrate a time-dependent entropy evolution that
aligns with the theoretical Page curve. This innovative experiment offers an alternative
framework for understanding the fundamental physics of black holes and validates entanglement
as a solution to the information paradox. The manuscript presents a compelling study with strong
theoretical underpinnings and innovative experimental design. By constructing a simplified
biphoton entanglement model, the authors effectively link fundamental concepts in quantum
mechanics to the astrophysical phenomenon of Hawking radiation. The manuscript is wellorganized,
and the results are clearly presented, with experimental findings validating theoretical
predictions.
This paper is a valuable contribution to the ongoing dialogue in theoretical physics and
cosmology, and I recommend it for publication after addressing the following points for revision:
1. The authors should review recent developments in gravity and black hole analogues such
as:
- "Optical simulations of gravitational effects in the Newton–Schrödinger
system." Nature Physics 11.10 (2015): 872-878.
- "Observation of a phase space horizon with surface gravity water waves."
Communications Physics 7.1 (2024): 165.
2. Can you briefly highlight in the introduction the broader implications of resolving the
information paradox in quantum gravity?
3. Can you please provide additional insights into how this model could be extended to
incorporate more complex phenomena, such as rotating or charged black holes?
4. Line 90 isn’t very clear, the definition minX{extX[]} needs to be clarified.
5. In lines 145-167, the full experimental apparatus is missing. What is the model of the
laser? What is the model of the BBO crystal? Powers? Duty cycle? Do you use type 1 or
type 2 SPDC?
6. In Figure. 2, you need to provide a full logical flow of the experiment in the caption.
7. Eq. 14 requires citation.
8. There are no units in Figure. 3 for time, also the axis are very small and the text has low
resolution.
9. The conclusion is very shot and is not informative. You could discuss how can you
extend this experiment. Or can it be measured in different gravity analogue apparatus
such as walking droplets, photonic structures or surface gravity waves?

Author Response

Dear Reviewer:

We sincerely appreciate the constructive feedback you provided. In response to your questions, we have addressed each point below and made the corresponding revisions in the article.

Question 1: The authors should review recent developments in gravity and black hole analogues such as:

    "Optical simulations of gravitational effects in the Newton–Schrödinger system." Nature Physics 11.10 (2015): 872-878.

    "Observation of a phase space horizon with surface gravity water waves."

Communications Physics 7.1 (2024): 165.

Response 1:

Thank you for your question.

We have inserted the following content between lines 45 - 48. 

However, recent advancements in gravity and black hole analogues—such as Bose-Einstein condensates [6], optical systems [14], and water waves [15]—have provided new insights into black hole dynamics.

Comments 2: Can you briefly highlight in the introduction the broader implications of resolving the information paradox in quantum gravity?

Response 2:

Thank you for your question.

We have inserted the following content between lines 29 - 36. 

The black hole information paradox reveals the potential conflict between general relativity and quantum mechanics under extreme gravitational conditions, involving fundamental issues such as quantum information conservation, black hole thermodynamics, and the second law of thermodynamics. Resolving this paradox not only constitutes a crucial test of black hole thermodynamics and the principle of quantum information conservation but also carries far-reaching implications for exploring new physical theories, deepening our understanding of the fundamental laws of the universe, and developing novel observational and experimental methodologies. ”

Comments 3: Can you please provide additional insights into how this model could be extended to incorporate more complex phenomena, such as rotating or charged black holes?

Response 3

Thank you for your question.

In the conclusion of the article, we further elaborated on this issue.

“ In future research, we plan to incorporate an ion–photon entanglement system to further expand the current experimental framework, thereby simulating more complex black hole physics scenarios such as rotating (Kerr) black holes, charged (Reissner - Nordstrom) black holes, and quantum gravity effects under higher-dimensional or dynamically evolving spacetimes etc.. In this ion–photon approach, the trapped ions can serve as the “black hole”, while the photons simulate the “Hawking radiation”. Their respective decoherence and entanglement recovery processes correspond to the apparent information loss and eventual restoration that occur during black hole evaporation. By encoding orbital angular momentum states or introducing effective charge interactions, it becomes possible to model the evolution of spin in a rotating black hole or charge in a charged black hole throughout the evaporation process, and to observe the dynamic transfer of quantum information between the radiated photons and the remaining system. Moreover, by combining high-dimensional entangled-state preparation, multi-modal quantum measurement techniques, and controllable noise channels, one can systematically investigate the impact of quantum fluctuations and spacetime quantization effects (e.g., “spacetime foam”) on information fidelity. The multi-degree-of-freedom controllability and high-precision manipulation afforded by ion–photon systems provide a highly flexible route for simulating the spin, charge, and complex spacetime structures of realistic black holes, offering valuable insights into the mechanisms of Hawking radiation and the black hole information paradox. 

Comments 4: Line 90 isn't very clear, the definition minX{extX[]} needs to be clarified.

Response 4:

Thank you for your question.

We further elaborated on this in lines 99 to 101 of the article.

“ ‘ext’ represents taking the extremum of the generalized entropy, and ‘min’ indicates that when there are multiple extremal surfaces, the one that minimizes S is chosen.

Comments 5: In lines 145-167, the full experimental apparatus is missing. What is the model of the laser? What is the model of the BBO crystal? Powers? Duty cycle? Do you use type 1 or type 2 SPDC?

Response 5

Thank you for your question.

The laser is a 40mW continuous wave laser with output polarized in the vertical direction. The BBO crystal consists of two thin BBO crystals with their optical axes perpendicular to each other, bonded together for the type-I parametric down-conversion process. Since a continuous-wave laser is used, the concept of a duty cycle does not apply. Here, the term is understood as the reception frequency of the signal light after parametric down-conversion, which is approximately 60 kHz.

The corresponding modifications and additions for this issue have also been made in lines 167 to 172 of the article.

Comments 6: In Figure. 2, you need to provide a full logical flow of the experiment in the caption.

Response 6

Thank you for your question.

We have made modifications to the caption of Figure 2:

Figure 2. Experimental light path diagram: the left module is optical path diagram of relative phase modulation and entangled state preparation, the right module is entanglement detection optical path diagram.(a) The pump laser (405 nm) is adjusted for polarization using wave plates before entering the BBO crystal to generate entangled photon pairs, which are then transmitted to paths A and B. (b) Each path contains a QWP and HWP for entanglement testing. A polarizer (PP) selects specific polarization states, and an optical coupler (OC) ensures signal transmission. The entangled photons eventually reach the single-photon detectors (SPD-A and SPD-B), with a coincidence counter (CCI) used to verify the photon entangled state.

Comments 7: Eq.14 requires citation.

Response 7

Thank you for your question.

This equation is from page 118 of reference [21] "PHOTONIC STATE TOMOGRAPHY" and has already been cited in the article.

Comments 8: There are no units in Figure. 3 for time, also the axis are very small and the text has low resolution .

Response 8

Thank you for your question.

The total time of the entire radiation process is set to 1, and the time axis represents the relative time at which the photon is radiated within the process, without specific units. The Figure has been adjusted according to the requirements.

Comments 9: The conclusion is very shot and is not informative. You could discuss how can you extend this experiment. Or can it be measured in different gravity analogue apparatus such as walking droplets, photonic structures or surface gravity wave.

Response 9

Thank you for your question.

We have made revisions to the conclusion and discussed how to extend this experiment. You can see the revised version in the conclusion section of the article.

We thank you for the opportunity to revise our manuscript.

Reviewer 2 Report

Comments and Suggestions for Authors

The paper by Li et. al. looks at a toy model for black hole evaporation and implements a simulation of this using an optics experiment. 

In its present form, I cannot recommend the paper for publication in Entropy. I list some of the main reasons below.

- The simulation of Hawking radiation has been long studied for example by P.D. Nation and M. Blencowe NJP (2010). There, a clear physical picture is presented in the Hamiltonian formalism, where the entanglement between infalling and outgoing radiation is clearly understood to be mode-mode entanglement (i.e. between signal and idler in an SPDC process). In the present paper there is no "phenomenological" process that is being investigated as a simulation of actual Hawking radiation. The authors simply state (or re-derive) the well known time dependence of von Neumann entropy and design their experiment to replicate this dependence "by hand."

- Related to the point above, the authors use entanglement between discrete variables as the model for the evolution of entanglement entropy in the evaporation process. I do not think this a very illuminating picture, since the minimal toy model should capture the fact that Hawking radiation arises due to the two-mode squeezed vacuum structure of the quantum field which gives rise to thermally when tracing over the modes trapped inside the horizon. This raises the question as to what extent the performed experiment can even be considered to be a simulation of the evaporation process. 

- For a relatively simple idea, the presentation of the paper is rather dense, especially the explanation of the theory section (Sec. II). Similarly, the explanation and description of the results of the experiment. Certain parts should be clarified and shortened, while others, such as the actual description of the experiment, should be expanded upon and a more concrete connection to the physics of the evaporation process can be made (currently it is just stated that separable photons are emitted, then at the halfway point entangled photons are used -- but this is just manually fixing what the authors want to happen in replicating the Page curve).

Comments on the Quality of English Language

The language can be improved in general, and is difficult to read at points. 

Author Response

Dear Reviewer:

We sincerely appreciate the constructive comments you have provided. In response to your inquiries, we address them point by point below, emphasizing the theoretical underpinnings of our model, the rationale behind the experimental design, and its connection to the physics of black hole evaporation.

Comments 1: The simulation of Hawking radiation has been long studied for example by P.D. Nation and M. Blencowe NJP (2010). There, a clear physical picture is presented in the Hamiltonian formalism, where the entanglement between infalling and outgoing radiation is clearly understood to be mode-mode entanglement (i.e. between signal and idler in an SPDC process). In the present paper there is no "phenomenological" process that is being investigated as a simulation of actual Hawking radiation. The authors simply state (or re-derive) the well known time dependence of von Neumann entropy and design their experiment to replicate this dependence "by hand."

Response 1:

Thank you for your question.

The reference you mentioned (Nation & Blencowe, 2010) describes a method for simulating black hole Hawking radiation using a system based on spontaneous parametric down-conversion (SPDC). In that approach, a thermal Hawking radiation spectrum is constructed through two-mode squeezed vacuum states. Grounded largely in quantum field theory, the study focuses on mode–mode entanglement in Hawking radiation and employs a Hamiltonian framework to analyze its evolution, which makes it highly suitable for describing the microscopic mechanism of black hole radiation.

By contrast, our work adopts a simplified toy model. In our model, the entangled photon pairs are likewise generated via SPDC. Tracing out one photon in each pair yields a single-photon state that is a mixed state, satisfying the condition for simulating the “thermal” properties of Hawking radiation. Our experiment proceeds by employing both this single-photon state and maximally entangled photon pairs, with particular attention paid to the evolution of von Neumann entropy during the Hawking radiation process, rather than replicating the details of quantum field modes.

In our experimental scheme, the “phenomenon” is simulated as follows: photons are treated as carriers of information, and we assume multiple pairs of entangled photons (A1–B1, A2–B2, …, An–Bn) are initially inside the black hole, with the number of photons corresponding to the black hole’s mass. Before the onset of Hawking radiation, the von Neumann entropy is zero because the system is in a pure state and the black hole mass is undiminished. As photons gradually escape (the A photons preceding the B photons), the black hole mass decreases to zero, while the von Neumann entropy first increases to a maximum and then decreases back to zero. This matches the entropy evolution predicted by Page and confirms the core prediction of the Page curve—that information is restored after the black hole has completely evaporated.

Regarding your concern about the “by hand” aspect of this “phenomenon,” we have elaborated on and emphasized it in lines 184–200 of our manuscript: “In this experiment, the black hole is … an entanglement fidelity of F > 0.977. As evaporation proceeds, all photons are ultimately radiated away.

The novelty of this method is reflected in the following:

Experimental feasibility: Compared with more complex field-theoretic simulations, our approach is straightforward to implement and leverages mature quantum technologies based on discrete entangled photon pairs. This simplification allows for more controlled experimental conditions and facilitates broader tests related to quantum gravity theories.

Intuitive information-recovery mechanism: By experimentally reproducing the entropy evolution trajectory, we provide controlled experimental support for the Page curve, rather than relying solely on numerical simulations.

Our approach is not intended as a substitute for field-theoretic methods; instead, we aim to provide experimental support for the universality of the Page mechanism from an information-theoretic perspective. Although our simulation is simplified, it preserves the key entanglement features of Hawking radiation and offers a novel experimental technique for testing potential resolutions of the black hole information paradox.

Comments 2: Related to the point above, the authors use entanglement between discrete variables as the model for the evolution of entanglement entropy in the evaporation process. I do not think this a very illuminating picture, since the minimal toy model should capture the fact that Hawking radiation arises due to the two-mode squeezed vacuum structure of the quantum field which gives rise to thermally when tracing over the modes trapped inside the horizon. This raises the question as to what extent the performed experiment can even be considered to be a simulation of the evaporation process.

Response 2:

Thank you for your question.

(1) Regarding your statement “since the minimal toy model should capture the fact that Hawking radiation arises due to the two-mode squeezed vacuum structure of the quantum field, which gives rise to thermal behavior when tracing over the modes trapped inside the horizon,” we would like to clarify as follows.

As noted in our response to the first question, because our primary interest lies in the evolution of von Neumann entropy during the Hawking radiation process, our focus is directed toward the entangled photon pairs generated after the SPDC process, rather than on the SPDC process itself. By tracing out one photon from these entangled pairs, we obtain a mixed single-photon state, which effectively captures the “thermal” characteristics of Hawking radiation. A detailed explanation is provided below:

[For detailed information on the formulas, please refer to the PDF file.]

(2) On the implicit question of whether discrete entanglement can be used as such simulation, our response is as follows.

The information-theoretic features of Hawking radiation (e.g., changes in von Neumann entropy) primarily depend on the formation and transfer of entanglement, rather than on the specific details of continuous field modes. The crux of information recovery is how the von Neumann entropy evolves over time. While continuous field modes may incorporate additional dynamic details, from an information-theoretic perspective, what truly matters is whether the model can track von Neumann entropy as it transitions from a pure state to a maximally mixed state and then returns to pure state. This evolution is key to validating the Page curve.

Hence, although a discrete system of entangled photons represents a minimal model and cannot replicate all the intricacies of a continuous field mode, it does capture the core mechanism of Hawking radiation and effectively conveys the primary features underlying information recovery. This does not impede our investigation into the essence of how information is restored. In our experiment, we successfully reproduce this trajectory by controlling the sequence in which photons “escape”.

In summary, we believe that the model presented in this work can fully serve to simulate the black hole evaporation process.

Comments 3: For a relatively simple idea, the presentation of the paper is rather dense, especially the explanation of the theory section (Sec. II). Similarly, the explanation and description of the results of the experiment. Certain parts should be clarified and shortened, while others, such as the actual description of the experiment, should be expanded upon and a more concrete connection to the physics of the evaporation process can be made (currently it is just stated that separable photons are emitted, then at the halfway point entangled photons are used -- but this is just manually fixing what the authors want to happen in replicating the Page curve).

Response 3:

Thank you for your question.

We have accepted the reviewer’s suggestions for improving our presentation and have made the following revisions.

Experiment Section.

In lines 166–172 (“The adjusted laser then…together for the type-I parametric down-conversion process.”), we have added specific details such as the model of the instruments used.

In the caption of Figure 2 (“(a) The pump laser (405 nm) is adjusted…used to verify the photon entangled state.”), we have included the experimental workflow.

Result Section.

In lines 261–274 (“This consistency demonstrates the reliability…simulate the resolution of the black hole information paradox.”), we have provided a more detailed analysis of the resulting curves, thereby strengthening the link between our experimental outcomes, the evaporation process, and information transfer.

Conclusion Section.

In lines 291–305 (“The essence of the information paradox lies in…platform for simulating the processes underlying black hole radiation.”), we have reiterated the connection between our model and experimental results and the black hole evaporation process and information transfer, and we have presented our concluding remarks.

Our experimental results provide a viable demonstration that entanglement dynamics can address the information paradox, in agreement with the predictions of the Page curve. Although the model is simplified, it effectively captures the key features of von Neumann entropy evolution during black hole evaporation, thus forging a closer connection between theory and experiment. We believe this work lays an important foundation for testing quantum gravity conjectures in a laboratory setting.

We thank you for the opportunity to revise our manuscript.

Reviewer 3 Report

Comments and Suggestions for Authors

This is an unusually fine scientific work offering a solution of the quantum information paradox related to black holes. It presents a further development of a proposal presented by Don Page some years ago. The work is well presented, and the results are interesting. In my opinion the paper needs no changes and can be published as it is. 

Author Response

Dear Reviewer,

Thank you for your kind and encouraging comments regarding our manuscript. We sincerely appreciate your time and effort in reviewing our work, as well as your positive assessment of its scientific contribution. We are especially grateful for your recognition of the way our study builds upon.

We are delighted that you found the paper well-presented and believe it merits publication without further changes. Your support and feedback motivate us to continue exploring this line of research.

Once again, thank you for your thorough review and recommendation.

Round 2

Reviewer 1 Report

Comments and Suggestions for Authors

I have carefully reviewed the revised version of the manuscript, and I am pleased to confirm that all previously suggested improvements and recommendations have been successfully implemented. The authors have provided a thorough and well-structured presentation of their work, with a clear theoretical framework, a well-designed experimental setup, and strong agreement between their experimental results and theoretical predictions.

The manuscript now convincingly demonstrates the viability of using biphoton entanglement as a model for simulating the evolution of von Neumann entropy in black hole Hawking radiation.

Given these considerations, I find that the manuscript is now suitable for publication in its current form. I recommend acceptance without further revisions.

Author Response

Dear Reviewer,

Thank you very much for your thorough review and positive feedback on our revised manuscript. We greatly appreciate the time and effort you have invested in evaluating our work. Your insightful comments and suggestions were invaluable in helping us refine our theoretical framework and experimental setup.

We are delighted to hear that our revisions have effectively addressed all your previous recommendations, and we appreciate your endorsement of our manuscript for publication in its current form. Your encouraging remarks regarding the clarity of our approach and the strong agreement between experimental results and theoretical predictions motivate us to continue our research in this field.

Thank you again for your support and for recommending our manuscript for acceptance.

Reviewer 2 Report

Comments and Suggestions for Authors

Overall the manuscript is improved and I am willing to reconsider my original recommendation. However there are still a few primary issues which the authors should address, as well as minor ones listed below. 

  • Overall, I think the authors do not make clear, or take time to address in sufficient detail, the role of analog gravity simulations in informing about unknown physics. The authors should at various key points in the manuscript (e.g. toward the end of the introduction, and in the conclusion) give their opinion as to what we learn about the resolution to the paradox from these experimental results. Obviously the key to the resolution is the purification of the localised state through leakage of photons out of the black hole, but does this say anything about what is needed (e.g. full quantum gravity, or simply semiclassical gravity?) for the resolution to the paradox. 
  • Likewise and related to the above, I think the authors should soften their language concerning what we do learn. In the conclusion, the authors state that their experiment demonstrates the Page curve, but this is fully explainable with known physics. They likewise say that the experiment “confirms” the prediction that radiation carries correlations allowing for the restoration of unitarity, however we still do not know how these correlations are carried away. 
  • Figure 1 is still not 100% clear to me (i.e. what it depicts and what the experiment simulates). The authors could clarify in particular what the role of the “optional surface” is, and how it relates to the purification of the  state / leakage of entangled partners, as well as what scenario they are implementing in their experiment with the 2-path SPDC process. 

Minor comments:

Line 43: The sentence should not start with “And”. Also, Angle need not be capitalised. 

Line 49: “Two-photon entangled system.”

Line 54: The authors should comment on what we learn from such analog systems in relation to the information paradox. For example, to what extent can one make the inference that unitarity is preserved in physical black holes? Or at least, the authors should comment on the limitations of what analog gravity can tell us about unknown physics (such as the resolution to the information paradox). 

Eq. 1. The constant refactor 1.48472 is not explained; this should be defined in terms of fundamental constants. 

Line 86: there is an issue with the formatting of “ref. (Equation 6.2) [18]” — maybe it should read Ref. [18] (see Equation 6.2). 

Line 109-131: I suggest the use of “phase” or something similar rather than “mode” since “mode” has a technical meaning it optics (and this is an optics paper). Indeed on page 19 they use the language of “phase” which should be made consistent with the description here. 

In Eq. (4), C_p and C_N are not defined

Line 183-199. What can be understood to be the physical mechanism for the photons being emitted through the black hole horizon? For example, is this to be interpreted purely semi classically, i.e. as in the works on the so-called “apparent horizon” in which the horizon either never forms [e.g. IJMPD 26 (12), 2017] or the horizon evaporates past the ingoing radiation [e.g. AJP 90 (2022)]. Or, do the authors favour a quantum-gravitational explanation [e.g. soft hair/asymptotic symmetries or nonlocal interactions across the horizon]? The authors should at least comment on this, because their phenomenological model purports to provide insight into the Page-curve turn over. In my understanding, the current setup would model the former case somewhat, insofar as the photons physically escape the black hole horizon and thus the horizon can only be interpreted as “apparent.”

Line 294: I would suggest using less strong language as “This process confirms that the seemingly thermal radiation actually carries quantum information…”

The authors have not confirmed anything like this, but rather than have engineered by hand a setup in which the partner modes are re-united with each other, thus signifying a purification of the state after the Page time. 

Line 316-322: Since the authors mention simulating higher-order quantum gravity effects in future works, perhaps they could mention relevant works in this direction [e.g. PRL 129, 181301 (2022), EJPC 82, 727 (2022)] that predict new “quantum gravitational effects” in toy model/analog systems.   

Comments on the Quality of English Language

Some minor typos can be fixed, mentioned above (though this list is not exhaustive). 

Author Response

Dear Reviewer:

We sincerely appreciate the constructive comments you have provided. In response to your inquiries, we address them point by point below.

Primary issues

Comments 1: Overall, I think the authors do not make clear, or take time to address in sufficient detail, the role of analog gravity simulations in informing about unknown physics. The authors should at various key points in the manuscript (e.g. toward the end of the introduction, and in the conclusion) give their opinion as to what we learn about the resolution to the paradox from these experimental results.

Response 1:

Thank you for your question. We have incorporated the reviewer’s suggestions to enhance our presentation and have made the following revisions:

We have added emphasis on how the simulated Hawking radiation implies the presence of unitarity in the black hole evaporation process—an essential component in potential solutions to the information paradox. At the end of the Introduction , in lines 50–55: “This study builds upon Page's theory by modeling the black hole as a Two-photon entangled system. Through a direct calculation of the von Neumann entropy, we derived and experimentally verified the Page curve, illustrating the time evolution of black hole entropy, which initially increases and subsequently decreases back to its initial value during Hawking radiation, thus successfully reproducing the key qualitative features of black hole evaporation and suggesting that unitarity may be preserved in physical black holes.”.

We have provided additional clarifications on the key points. In line 210-211: “This reconstruction is evidenced by the average coincidence count—indicating entanglement reestablishment—and an entanglement fidelity exceeding F > 0.977, which strongly suggest that the process is consistent with unitarity, as no irreversible loss of information is detected within the measured subsystem.”.

We supplemented the article according to the comments. In lines 284–285 : “This result is in agreement with Page's prediction, which indicates that the process preserves unitarity and also suggests that the dynamic separation mechanism of entangled photon pairs can effectively simulate the resolution of the black hole information paradox.”.

We revised the text and added discussions pertaining to the significance and limitations of our experiment. In the Conclusion: “The information paradox arises from the apparent contradiction between the purportedly thermal characteristics of Hawking radiation and the fundamental principle of quantum information conservation. Our experimental framework reveals that through controlled spatial separation and subsequent interference of entangled photon pairs, the quantum state of the radiation subsystem undergoes a transformation sequence: starting from an initial pure state (characterized by low entropy) to a mixed state (exhibiting elevated entropy), and ultimately recovering purity (returning to low entropy). This observation suggests that radiation possessing thermal-like properties may maintain quantum correlations with black hole's interior particles. Following complete spatial separation and subsequent recombination of all entangled pairs, the radiation subsystem's von Neumann entropy approaches initial value, indicating full information retention during the simulated black hole evaporation process. Therefore, the evolution is consistent with quantum mechanical unitarity principles and can be interpreted as the purification of the localized state via the leakage of photons, is central to addressing the black hole information paradox. However, since this evolution process does not provide a comprehensive and detailed simulation of the complete Hawking radiation process, its result should not be regarded as a definitive resolution of the black hole information paradox. To fully resolve the paradox, a more advanced experimental system and more rigorous theoretical support remain necessary. Nonetheless, this experiment clearly indicates that both the transfer of information and the reconstruction of entanglement during the process of black hole Hawking radiation can be effectively explained by fully quantum processes. This result not only deepens our understanding of the mechanisms underlying information conservation in black hole Hawking radiation but also indicates that entangled photon systems can serve as a promising platform for further exploring the fundamental processes in black hole thermodynamics.”.

Comments 2: Obviously the key to the resolution is the purification of the localised state through leakage of photons out of the black hole, but does this say anything about what is needed (e.g. full quantum gravity, or simply semiclassical gravity?) for the resolution to the paradox.

Response 2:

Thank you for your question. We agree that the key is indeed the purification of the local state.

We clarify that our model does not incorporate the scattering effects included in Page’s model of the radiation process. As a result, even though the evolution trend we obtain resembles that of the Page curve, the two do not fully coincide. In lines 287–289: “Since the biphoton entanglement model proposed in this paper cannot consider for the scattering effects included in the radiation process of the Page model, the 'transition time' of the two curves does not completely coincide.”

Furthermore, we note that due to the simplifications in our model, these results should not be taken as a definitive resolution of the black hole information paradox. A more comprehensive experimental setup and more robust theoretical support remain necessary for a complete resolution. In lines 317–321:“However, since this evolution process does not provide a comprehensive and detailed simulation of the complete Hawking radiation process, its result should not be regarded as a definitive resolution of the black hole information paradox. To fully resolve the paradox, a more advanced experimental system and more rigorous theoretical support remain necessary.”.

Accordingly, we mention our plan to introduce an Ion–photon entangled system, thereby incorporating additional degrees of freedom into the simulation for a more realistic black hole model. In the Conclusion, the lines 328–346: “In future research, we plan to incorporate an ion–photon entanglement system to further expand the current experimental framework, thereby simulating more complex black hole physics scenarios such as rotating (Kerr) black holes[25], charged (Reissner–Nordstrom) black holes[26], and quantum gravity effects[27, 28] under higher-dimensional or dynamically evolving spacetimes etc.. In this ion–photon approach, the trapped ions can serve as the “black hole”, while the photons simulate the “Hawking radiation”. Their respective decoherence and entanglement recovery processes correspond to the apparent information loss and eventual restoration that occur during black hole evaporation. By encoding orbital angular momentum states or introducing effective charge interactions, it becomes possible to model the evolution of spin in a rotating black hole or charge in a charged black hole throughout the evaporation process, and to observe the dynamic transfer of quantum information between the radiated photons and the remaining system. Moreover, by combining high-dimensional entangled-state preparation, multi-modal quantum measurement techniques, and controllable noise channels, one can systematically investigate the impact of quantum fluctuations and spacetime quantization effects (e.g., “spacetime foam”) on information fidelity. The multi-degree-of-freedom controllability and high-precision manipulation afforded by ion–photon systems provide a highly flexible route for simulating the spin, charge, and complex spacetime structures of realistic black holes, offering valuable insights into the mechanisms of Hawking radiation and the black hole information paradox.”. 

Because the current experimental simulation does not incorporate complete quantum gravitational effects—such as quantum fluctuations of the gravitational field or deeper quantum structures of spacetime—the present findings cannot definitively determine which theoretical framework (full quantum gravity versus semiclassical gravity) is ultimately required to resolve the information paradox. Nonetheless, our results indicate that the ‘entanglement/purification mechanism driven by photon leakage’ is consistent with existing models of black hole evaporation, thereby laying the groundwork for incorporating more quantum gravitational effects or more complex entanglement degrees of freedom in future research.

Comments 3: Likewise and related to the above, I think the authors should soften their language concerning what we do learn. In the conclusion, the authors state that their experiment demonstrates the Page curve, but this is fully explainable with known physics. They likewise say that the experiment “confirms” the prediction that radiation carries correlations allowing for the restoration of unitarity, however we still do not know how these correlations are carried away.

Response 3:

Thank you for your question. We fully agree with the reviewer’s suggestion to soften our wording. Accordingly, we have revised lines 302–327 of the main text to address this concern.

“The information paradox arises from the apparent contradiction between the purportedly thermal characteristics of Hawking radiation and the fundamental principle of quantum information conservation. Our experimental framework reveals that through controlled spatial separation and subsequent interference of entangled photon pairs, the quantum state of the radiation subsystem undergoes a transformation sequence: starting from an initial pure state (characterized by low entropy) to a mixed state (exhibiting elevated entropy), and ultimately recovering purity (returning to low entropy). This observation suggests that radiation possessing thermal-like properties may maintain quantum correlations with black hole's interior particles. Following complete spatial separation and subsequent recombination of all entangled pairs, the radiation subsystem's von Neumann entropy approaches initial value, indicating full information retention during the simulated black hole evaporation process. Therefore, the evolution is consistent with quantum mechanical unitarity principles and can be interpreted as the purification of the localized state via the leakage of photons, is central to addressing the black hole information paradox. However, since this evolution process does not provide a comprehensive and detailed simulation of the complete Hawking radiation process, its result should not be regarded as a definitive resolution of the black hole information paradox. To fully resolve the paradox, a more advanced experimental system and more rigorous theoretical support remain necessary. Nonetheless, this experiment clearly indicates that both the transfer of information and the reconstruction of entanglement during the process of black hole Hawking radiation can be effectively explained by fully quantum processes. This result not only deepens our understanding of the mechanisms underlying information conservation in black hole Hawking radiation but also indicates that entangled photon systems can serve as a promising platform for further exploring the fundamental processes in black hole thermodynamics.” 

Comments 4: Figure 1 is still not 100% clear to me (i.e. what it depicts and what the experiment simulates). The authors could clarify in particular what the role of the “optional surface” is, and how it relates to the purification of the  state / leakage of entangled partners, as well as what scenario they are implementing in their experiment with the 2-path SPDC process.  

Response 4:

Thank you for your question.

In fact, the pictures shown in Figures 1 are derived from an entanglement perspective based on Formula (2). This model differs from the “biphoton entanglement model” presented in this paper. The primary purpose of proposing this model is to verify the phenomenological and procedural validity of the “biphoton entanglement model”. For the reviewer’s convenience, I will explain the logic of the article starting from the figures:

 As described in lines 108–114 of the paper. In Figure 1(a), the center denotes the singularity, and the dashed line represents the optional surface X. The region between the singularity and X corresponds to the black hole portion in the generalized entropy approach, where the area entropy is used for calculation. The region between X and the cutoff boundary (the red line) is likewise treated as a separate region Sigma_X , whose von Neumann entropy is derived from the vacuum fluctuations of the quantum field. The blue line indicates the event horizon. Surface X can be placed anywhere between the singularity and the event horizon, while the region outside the cutoff boundary is not included in our calculation.

Before black hole evaporation begins, the vacuum quantum field does not contribute any entropy, and there are no particles within Sigma_X . Consequently, the minimum value is attained by placing X directly at the singularity, yielding a von Neumann entropy of zero. If one keeps X at the singularity as radiation commences, some radiated particles cross the cutoff boundary to reach the exterior of the black hole, whereas other particles remain in Sigma_X , causing the von Neumann entropy to rise. This situation corresponds to Figure 1(b), which is effectively equivalent to computing the von Neumann entropy of the radiated particles outside the black hole. This situation is defined as Mode 1, as detailed in lines 115–124 of the paper.

It becomes evident that, once radiation starts, the von Neumann entropy under Mode 1—where X remains at the singularity—may no longer be minimal, necessitating a discussion of the case where X is not at the singularity. If X is placed at the event horizon, as shown in Figure 1(c), Sigma_X contains only a small number of relatively independent radiated particles, and its von Neumann entropy is certainly not zero. At this stage, if X is moved too far toward the singularity, the area entropy will decrease, but the von Neumann entropy will rise, as illustrated in Figure 1(d). In contrast, if X is shifted only slightly toward the singularity, both the area entropy and the von Neumann entropy decrease (as shown in Figure 1(e)), which corresponds to an appropriate choice for X. This scenario, where X remains near the event horizon, is defined as Mode 2, whose minimal value approximates the black hole’s area entropy. As discussed in lines 125–142 of the paper.

The model equation (Formula (2)) underlying these figures have been proven in Reference [19]. Therefore, we can establish a one-to-one correspondence between the phenomena and processes depicted in these figures and the “biphoton entanglement model” proposed in this paper, as presented in Table 1 of the main text.

In summary, both Formula (2) and Figures 1 serve to validate the rationality of the “biphoton entanglement model” advanced in this paper. If one wishes for the optional surface X to be applied in subsequent experimental accounts, it may be interpreted as a phenomenological construct functioning as an operational partitioning tool, as explained in our response to Comments 9.

Minor comments:

Comments 5: Line 43: The sentence should not start with “And”. Also, Angle need not be capitalised.

Line 49: “Two-photon entangled system.”

Response 5:

Thank you for your question.

We have accepted the reviewer’s suggestions for improving our presentation and have made the following revisions.

We revised the article according to the comments of the reviewers. In lines 42–43: “Moreover, the Island rule formula of Hawking radiation fine entropy is put forward, and the Page curve is derived from the angle of pure gravity[11][12][13].”.

We revised the article according to the comments of the reviewers. In lines 50: “This study builds upon Page's theory by modeling the black hole as a Two-photon entangled system.”.

Comments 6: Line 54: The authors should comment on what we learn from such analog systems in relation to the information paradox. For example, to what extent can one make the inference that unitarity is preserved in physical black holes? Or at least, the authors should comment on the limitations of what analog gravity can tell us about unknown physics (such as the resolution to the information paradox).

Response 6:

Thank you for your question.

We have accepted the reviewer’s suggestions for improving our presentation and have made the following revisions。

We supplemented and revised the article according to the comments of the reviewers. In lines 50-59: “Through a direct calculation of the von Neumann entropy, we derived and experimentally verified the Page curve, illustrating the time evolution of black hole entropy, which initially increases and subsequently decreases back to its initial value during Hawking radiation, thus successfully reproducing the key qualitative features of black hole evaporation and suggesting that unitarity may be preserved in physical black holes. Although the simplified model does not fully capture the complex dynamics of real black holes and has certain limitations in addressing unknown physical phenomena, it offers a valuable perspective for intuitively understanding the black hole information paradox and the process of information transfer, thereby laying a foundation for further research in this field.”.

Comments 7: Eq. 1. The constant refactor 1.48472 is not explained; this should be defined in terms of fundamental constants.

Line 86: there is an issue with the formatting of “ref. (Equation 6.2) [18]” — maybe it should read Ref. [18] (see Equation 6.2). .

Response 7:

Thank you for your question.

We have accepted the reviewer’s suggestions for improving our presentation and have made the following revisions

We supplemented the article according to the comments of the reviewers. In lines 77-79: “The 1.48472 is the ratio of the increasing entropy of the radiating part to the decreasing entropy of the black hole part.”.

We revised the article according to the comments of the reviewers. In lines 92-93: “According to the von Neumann entropy formula for black holes given in Ref.[19](see Equation 6.2).”.

Comments 8: Line 109-131: I suggest the use of “phase” or something similar rather than “mode” since “mode” has a technical meaning it optics (and this is an optics paper). Indeed on page 19 they use the language of “phase” which should be made consistent with the description here.

In Eq. (4), C_p and C_N are not defined .

Response 8:

Thank you for your question.

In fact, the term 'Mode' here does not have the same meaning as 'phase' in lines 280-283 on page 10. 'Mode' refers to the selection method of the optional surface X (as shown in Table 1), whereas 'phase' represents different stages in the simulation of the Hawking radiation process. Although 'Mode' has another meaning in optics (such as the spatial distribution patterns of laser modes), we believe that continuing to use 'Mode' in this context is more appropriate for the explanation in this paper.

A detailed introduction to C_p and C_N in Eq. (4) can be found in Appendix A, lines 383-388 (A16-A19), where the derivation process is also provided.

Comments 9: Line 183-199. What can be understood to be the physical mechanism for the photons being emitted through the black hole horizon? For example, is this to be interpreted purely semi classically, i.e. as in the works on the so-called “apparent horizon” in which the horizon either never forms [e.g. IJMPD 26 (12), 2017] or the horizon evaporates past the ingoing radiation [e.g. AJP 90 (2022)]. Or, do the authors favour a quantum-gravitational explanation [e.g. soft hair/asymptotic symmetries or nonlocal interactions across the horizon]? The authors should at least comment on this, because their phenomenological model purports to provide insight into the Page-curve turn over. In my understanding, the current setup would model the former case somewhat, insofar as the photons physically escape the black hole horizon and thus the horizon can only be interpreted as “apparent.”

Response 9:

Thank you for your question.

We have accepted the reviewer’s suggestions for improving our presentation and have made the following revisions.

We supplemented the article according to the comments of the reviewers. In lines 190-195: “The black hole evaporation process is entirely framed within the context of quantum optics theory, with the “horizon”—treated as a phenomenological construct corresponding to the optional surface X-serves as an operational partitioning tool. This tool divides the observable radiation (the extracted photons) from the simulated black hole microstates (the remaining photons) to characterize the entanglement dynamics between the radiated and residual components.”.

The reviewer's interpretation—“In my understanding, ... only be interpreted as “apparent.””—is not entirely correct. In this paper, our model is constructed within the theoretical framework of quantum optics, without considering quantum gravity theory. Therefore, it differs from the reviewer's understanding of “a quantum-gravitational ... across the horizon]”.

In the experiment, the “horizon” essentially corresponds to the optional surface X in the preceding model. However, it does not possess “geometric properties” or any spacetime causal structure. Strictly speaking, it is merely a phenomenological “operational” concept or classification tool, used to separate the observable radiation (extracted photons) from the simulated microscopic black hole states (remaining photons). Therefore, it may not be appropriate to describe it using a semiclassical interpretation.

Comments 10: Line 294: I would suggest using less strong language as “This process confirms that the seemingly thermal radiation actually carries quantum information…”

The authors have not confirmed anything like this, but rather than have engineered by hand a setup in which the partner modes are re-united with each other, thus signifying a purification of the state after the Page time.

Line 316-322: Since the authors mention simulating higher-order quantum gravity effects in future works, perhaps they could mention relevant works in this direction [e.g. PRL 129, 181301 (2022), EJPC 82, 727 (2022)] that predict new “quantum gravitational effects” in toy model/analog systems.  

Response 10:

Thank you for your question.

We have accepted the reviewer’s suggestions for improving our presentation and have made the following revisions.

We revised the article according to the comments of the reviewers as show in the response of comment 3.

We supplemented the article according to the comments of the reviewers. In lines 329-331: “In future research, we plan to incorporate an ion–photon entanglement system to further expand the current experimental framework, thereby simulating more complex black hole physics scenarios such as rotating (Kerr) black holes[25], charged (Reissner–Nordstrom) black holes[26], and quantum gravity effects[27,28] under higher-dimensional or dynamically evolving spacetimes etc..”.

We thank you for the opportunity to revise our manuscript.

Round 3

Reviewer 2 Report

Comments and Suggestions for Authors

I think the authors have done a satisfactory job in improving the manuscript, particularly in the expanded exposition on the contribution of their experiment to the broader literature on simulations of the Hawking process. While I still think that the results are "obvious" insofar as there is no other option except for unitarity to be preserved (since there is no irretrievable loss of any photons), I think it may serve as a foundation for deeper explorations, as the authors suggest they are planning to carry out in the future directions paragraph of their conclusion. 

Minor typos:

Line 7: “may contribute…”

Line 147: there is a space between “black hole” and the comma

Line 260: “As the Hawking process progresses, the number of radiation particles will increase with time…”